# Hemodynamic Effect of Pulsatile on Blood Flow Distribution with VA ECMO: A Numerical Study

**DOI:** 10.3390/bioengineering9100487

**Published:** 2022-09-20

**Authors:** Kaiyun Gu, Sizhe Gao, Zhe Zhang, Bingyang Ji, Yu Chang

**Affiliations:** 1National Clinical Research Center for Child Health, The Children’s Hospital, Zhejiang University School of Medicine, Hangzhou 310005, China; 2Department of Cardiopulmonary Bypass, National Center for Cardiovascular Disease, Chinese Academy of Medical Sciences & Peking Union Medical College Fuwai Hospital, Beijing 100037, China; 3Cardiology Department, Peking University Third Hospital, Beijing 100191, China

**Keywords:** hemodynamics, veno-arterial extracorporeal membrane oxygenation (VA ECMO), intra-aortic balloon pump (IABP), pulsatility, numerical analyses

## Abstract

The pulsatile properties of arterial flow and pressure have been thought to be important. Nevertheless, a gap still exists in the hemodynamic effect of pulsatile flow in improving blood flow distribution of veno-arterial extracorporeal membrane oxygenation (VA ECMO) supported by the circulatory system. The finite-element models, consisting of the aorta, VA ECMO, and intra-aortic balloon pump (IABP) are proposed for fluid-structure interaction calculation of the mechanical response. Group A is cardiogenic shock with 1.5 L/min of cardiac output. Group B is cardiogenic shock with VA ECMO. Group C is added to IABP based on Group B. The sum of the blood flow of cardiac output and VA ECMO remains constant at 4.5 L/min in Group B and Group C. With the recovery of the left ventricular, the flow of VA ECMO declines, and the effective blood of IABP increases. IABP plays the function of balancing blood flow between left arteria femoralis and right arteria femoralis compared with VA ECMO only. The difference of the equivalent energy pressure (dEEP) is crossed at 2.0 L/min to 1.5 L/min of VA ECMO. PPI’ (the revised pulse pressure index) with IABP is twice as much as without IABP. The intersection with two opposing blood generates the region of the aortic arch for the VA ECMO (Group B). In contrast to the VA ECMO, the blood intersection appears from the descending aorta to the renal artery with VA ECMO and IABP. The maximum time-averaged wall shear stress (TAWSS) of the renal artery is a significant difference with or not IABP (VA ECMO: 2.02 vs. 1.98 vs. 2.37 vs. 2.61 vs. 2.86 Pa; VA ECMO and IABP: 8.02 vs. 6.99 vs. 6.62 vs. 6.30 vs. 5.83 Pa). In conclusion, with the recovery of the left ventricle, the flow of VA ECMO declines and the effective blood of IABP increases. The difference between the equivalent energy pressure (EEP) and the surplus hemodynamic energy (SHE) indicates the loss of pulsation from the left ventricular to VA ECMO. 2.0 L/min to 1.5 L/min of VA ECMO showing a similar hemodynamic energy loss with the weak influence of IABP.

## 1. Introduction

Veno-arterial (VA) ECMO is a rescue therapy that can stabilize hemodynamically compromised patients, the primary indications of which are multifactorial cardiogenic shock (CS) and cardiac arrest [1,2]. The ECMO circuit draws deoxygenated blood from the venous system with a non-pulsatile pump through an oxygenator, where gas is exchanged, and blood is returned to the systemic circulation [3]. Clinically, VA ECMO has been proven to be beneficial to patients diagnosed with CS that originates from cardiovascular disorders, including acute myocardial infarction and myocarditis [4,5]. However, the benefits of ECMO should be weighed against inherent risks, including coagulation disorders, inflammatory responses, elevated total peripheral resistance, and intravascular sludging that may negatively impact microcirculation [6]. In addition, the retrograde continuous flow of ECMO impairs the pulsatility of native cardiac ejection, further weakening circulatory pulsatility and reducing end-organ perfusion as well as tissue metabolism [7].

The pulsatile properties of arterial flow and pressure are important. At present, the pulse pressure index (PPI), calculated as the ratio of pulse pressure to the mean arterial pressure, defines pulsatility [8]. Although hemodynamic energy is reduced with decreasing pulsatile frequency at the same flow rate, energy equivalent pressure (EEP) and surplus hemodynamic energy (SHE) are applied to evaluate pulsatility quantitatively [9]. It has been reported that low pulsatility led to a series of adverse events. Long-term reduced pulsatile flow leads to ischemic stroke [10], pump embolism [11], and increased aortic vascular stiffness [12]. Blood viscoelasticity exists when there is low pulsatile flow, which increases resistance and directly impacts organ perfusion. On the contrary, with an increase in pulsatility, the viscoelasticity reduces, and oxygen delivery can be improved [13,14,15]. Although Russel reported that the long-term impact of non-pulsatile blood flow was not associated with worsening end-organ perfusion [16], Leslie commented that it increased the rate of gastrointestinal bleeding [17]. Plonek et al. proved that myocardial contraction influences stress in the aortic wall and impacts longitudinal systolic aortic stretching [18]. Pulsatile blood flow is crucial for maintaining organ perfusion as well as microcirculation, and decreased pulsatility changes the blood flow of the circulatory system, affecting organ perfusion and blood vessel reconstruction.

Inserted in the descending aorta, the intra-aortic balloon pump (IABP) is one of the most commonly used circulatory supports. IABP can increase diastolic blood pressure, decrease afterload as well as myocardial oxygen consumption, increase coronary artery perfusion, and moderately enhance cardiac output [19,20,21]. In the 2000s, Onorati’s group focused on the pulsatile property generated by IABP during cardiopulmonary bypass (CPB), and their studies proved the superiority of pulsatile flow over non-pulsatile flow [22,23]. IABP is no longer routinely recommended in patients diagnosed with CS complicated by acute myocardial infarction [24,25] because the IABP-SHOCK II trial revealed that it did not improve their short and long-term outcomes [26]. However, when IABP is used concomitantly with VA-ECMO, its counter pulsatile flow helps systolic unloading and enhances the myocardial oxygen supply-demand ratio, improving clinical outcomes [27,28]. Several studies have reported that the combined application of VA ECMO and IABP had a higher weaning rate than VA ECMO alone [28,29,30]. In a recent meta-analysis, Li et al. reported that concomitant initiation of IABP in combination with VA-ECMO in CS patients was associated with a lower in-hospital mortality rate [31]. Above all, the improved outcomes of combining IABP and VA ECMO partly relied on the intrinsic pulsatility of IABP, which increased the pulsatile property of blood flow. Nevertheless, the hemodynamic effect of pulsatile blood flow in improving the blood flow distribution of VA ECMO supported by the circulatory system remains unclear.

The purpose of this research is to analyze the hemodynamic effect of pulsatile flow in improving blood flow distribution of veno-arterial extracorporeal membrane oxygenation (V-A ECMO) supported circumstances. Finite element models consisting of the aorta, VA ECMO, and IABP, calculated the fluid-structure interaction (FSI) of the mechanical response. CS, VA ECMO, and VA ECMO plus IABP models were used to explore the effect of different support types on the blood supply to the coronary artery, brain, and lower limb to correlate between heart flow and ECMO flow, and vascular wall stress. For the three groups studied, Group A consisted of CS patients having 1.5 L/min of cardiac output (CO); Group B consisted of CS patients with VA ECMO; in Group C, IABP patients were included in the Group B configuration, to generate pulsatile blood flow. The sum of the blood flow of CO and VA ECMO remained constant at 4.5 L/min in Group B and Group C. EEP, SHE, PPI, and the perfusion flow of the coronary, brain, and kidney arteries were extracted. In addition, the flow intersection and the distribution of wall shear stress (WSS) were selected as parameters to evaluate the hemodynamic influence.

## 2. Materials and Methods

### 2.1. Reconstruction of Aortic, ECMO, and IABP Model

In this study, an aortic geometric model of a CS patient in combination with VA ECMO and IABP was reconstructed based on spiral computer tomography (CT) image data. The study was approved by the ethical review board (Number: IRB 00006761-M2018272). MIMICS (Materialize, Leuven, Belgium), innovation suite software, and Geomagic (Geomagic, NC, USA), a professional engineering software, were used for the reconstruction and smoothing of the three-dimensional aortic model.

One cannula of VA ECMO withdraws deoxygenated blood from the venous system, and the other cannula is implanted into the arteria femoralis to return the oxygenated blood to systemic circulation. In this study, the arterial catheter was used as the blood inlet to implant the arteria femoralis in Figure 1. The IABP with a 30 mL balloon was modeled to supply pulsatile blood flow, which was implanted into the descending aorta. The IABP is counter-pulsatile to the ejection of the left ventricle. Specifically, LV systole encounters the balloon-off, whereas LV diastole encounters the balloon-on.

### 2.2. FE Model

Finite element (FE) meshes were generated for the aortic model using Hypermesh (Altair Engineering, MI), and 3D solid elements (hexahedral elements) were used to capture the stress/displacement. The fluid elements were used to obtain the distribution of blood flow. Grid-independence and cycle-independence are sufficient for the requirements based on our previous research [8]. The results demonstrate that 347,098 and 58,509 hexahedral elements are sufficient for this study.

For this study, the constitutive model was used to simulate the mechanical response of the aortic tissue. The anisotropic Mooney–Rivlin model was adopted to characterize the aortic mechanisms. The strain energy function is expressed as follows:(1)W=c1(I1−3)+c2(I2−3)+D1e[D2(I1−3)−1]+k1k2e[k2(I4−1)2−1]
where $I_{1}$ is the first invariant of the deviatoric part of the right Cauchy Green deformation tensor; $I_{2}$ is the second invariant of the deviatoric part of the right Cauchy Green deformation tensor; $I_{4}$ is the unit vector in the circumferential direction of the vessel; and $c_{1}$, $c_{2}$, $D_{1}$, $D_{2}$, $k_{1}$, $k_{2}$ are material parameters [32].

For blood flow, Navier–Stokes equations with arbitrary Lagrangian-Eulerian (ALE) formulation were used to solve the governing equations. The blood flow characteristics were assumed to be Laminar, Newtonian, viscous, adiabatic, and incompressible. The blood density and viscosity were 1050 kg/m^3^ and 0.0035 Pa·s [8], respectively.

### 2.3. Boundary Condition

To analyze the hemodynamic effect of pulsatile blood flow distribution in the VA ECMO-supported circulatory system, three groups, including CS, VA ECMO, and VA ECMO plus IABP, were investigated in our study (Table 1). Group A consisted of CS patients with 1.5 L/min of cardiac output (CO), which is “native CO” from the echocardiographs of the flow velocity of the aortic valve. The calculation formula is $\mathrm{CO}=\int Vdt*A_{av}*HR$ where *V* is the flow velocity of the aortic valve (Figure 2A). *A*_av_ is the area of the aortic valve, which is 715.44 mm^2^. HR is heart rate (75 bpm); Group B consisted of CS patients with VA ECMO, who were in the process of cardiac recovery; In Group C patients, IABP was connected to the configuration of Group B to generate the pulsatile blood flow. The sum of the blood flow during CO and VA ECMO remained constant at 4.5 L/min in groups B and C, respectively. The flow velocity of the aortic valve as the inlet boundary was extracted from the echocardiographs of the patient to calculate the CO. The flow velocities of the aortic valve and VA ECMO are shown in Figure 2. The mean arterial pressure (MAP) of 70 mm Hg as the outlet boundary was set for the whole vasculature to demonstrate the systemic afterload. The balloon radius of IABP as the parameters was controlled. In the systolic period, the IABP balloon was at its minimum volume. After the aortic valve closed, IABP began to inflate. Then, the balloon was vented after reaching 30 mL [8]. The balance of cardiac function, ECMO blood flow, and the pulsatile effect under the support of VA ECMO was analyzed based on the distribution of the circulatory system.

Numerical analyses were performed using ADINA (ADINA R & D, Watertown, MA, USA), which is suitable for advanced material models and FSI. The heart rate was approximately 75 bpm, which corresponded to a cardiac cycle of 0.8 s. The magnitude of the temporal resolution was set at 8 ms to perform the numerical calculation. The cardiac cycle started at the systole phase.

### 2.4. Biomechanical Analysis

Three indicators, energy equivalent pressure (EEP), surplus hemodynamic energy (SHE), and pulse pressure index (PPI), were calculated to evaluate the effects of pulsatile VA ECMO on the distribution of the circulatory system.

The Energy Equivalent Pressure (EEP) is the ratio between the integral of the hemodynamic power and the integral of the flow during each pulse cycle [6].
(2)EEP (mmHg)=(∫fpdt)/(∫fdt)
(3)$$\mathrm{dEEP}=(\int f_{lv}p_{lv}dt)/\left(\int f_{lv}dt\right)-\left(\int f_{ecmo}p_{ecmo}dt\right)/\left(f_{ecmo}dt\right)$$
where f is the flow velocity of the aortic valve, p is the arterial pressure, and dt indicates that integration is performed over time (t). $f_{lv},p_{lv}$ represents the flow and pressure of the LV. $f_{ecmo},p_{ecmo}$ refers to the flow and pressure of the VA ECMO catheter. dEEP is the difference between LV EEP and VA ECMO EEP.

The surplus hemodynamic energy (SHE) was calculated by multiplying the difference between EEP and MAP by 1332. The SHE is equal to the extra energy in terms of energy units [6].
(4)$$\mathrm{SHE}\;({\mathrm{ergs}/\mathrm{cm}}^{3})=1332\left[\left(\int fpdt\right)/\left(\int fdt\right)-MAP\right]$$
(5)$$\mathrm{dSHE}=1332\left[\left(\left(\int f_{lv}p_{lv}dt\right)/\left(\int f_{lv}dt\right)-MAP)-(\left(\int f_{ecmo}p_{ecmo}dt\right)/\left(f_{ecmo}dt\right)-MAP\right)\right]$$
where dSHE is the difference value between LV SHE and VA ECMO SHE.

The pulse pressure index (PPI), another parameter used to assess pulsatility, was calculated by Equation (6) [33]. However, the curve of diastolic pressure is changed with the IABP support because of the pulsating pressure in the diastole period. The PPI method was revised to fit the calculation of the pulse using Equation (7).
(6)$$\mathrm{PPI}=\frac{\left(SBP-DBP\right)}{MAP}$$
(7)$${\mathrm{PPI}}^{\prime }=\frac{\left(SBP-DBP\right)+(DBP_{peak}-DBP)}{\int pdt}$$
where SBP is the systolic blood pressure, and DBP is the diastolic blood pressure. $DBP_{peak}$ is the peak diastolic pressure. SBP, DBP, and $DBP_{peak}$ were marked in the curve of Figure 2C, which is under the IABP support. p is arterial pressure. ${\mathrm{PPI}}^{\prime }$ is the revised pulse pressure index. When IABP-off, the value of $DBP_{peak}-DBP$ was zero.

## 3. Results

To evaluate the hemodynamic effect of pulsatile blood flow on the blood distribution of VA ECMO, EEP, SHE, PPI, the perfusion flow of the coronary, brain, and kidney, arteries, flow intersection, and the distribution of vessel shear stress were assessed and are illustrated in Figure 3, Figure 4, Figure 5, Figure 6 and Figure 7. Two time points (T1 and T2) were selected (Figure 7). T1 is the maximum value of blood velocity during the systole period, and T2 is the maximum volume of the IABP during the diastole period.

Figure 3 shows the total blood flow of the coronary artery, brain, kidneys, and lower limbs. The sum of the inlet flow, including CO and VA ECMO, was 4.5 L/min. In the VA ECMO group (Group B), the maximum outlet flow was 4.72 L/min because of the elastic characteristics of the aorta. The relative error was less than 5%, which was sufficient for this study. For the VA ECMO and IABP groups, a complete superposition effect was not observed under VA ECMO flow and IABP with 30 mL per cardiac cycle. The effective blood flow of the IABP was calculated to contradict VA ECMO, which is the increase in total flow. The minimum effective blood flow of IABP for the 3 L/min of VA ECMO is 0.78 L min. With the recovery of the left ventricle, the VA ECMO flow decreased, and the effective blood flow of IABP increased.

Figure 4 shows the distribution of the arteria femoralis. VA ECMO catheter was implanted in the left arteria femoralis (LAF). A significant decrease was observed in the LAF flow compared with the right arteria femoralis (RAF) in either VA ECMO or VA ECMO and IABP. With recovered cardiac function and decreased VA ECMO flow, LAF and RAF flow gradually increased with VA ECMO support only. However, the flow of the arteria femoralis showed a different trend from that of VA ECMO and IABP. The flow of LAF (ECMO and IABP) showed an upward trend, but RAF (ECMO & IABP) showed a downward trend with the decline in VA ECMO flow. Contrast with the flow of RAF and RAF (ECMO and IABP), is contrary to the trend. IABP balanced the blood flow as compared with that of VA ECMO alone.

Figure 5 shows arterial pulsatile properties. dSHE is the difference between LV SHE and VA ECMO SHE, and dEEP is the difference between LV EEP and VA ECMO EEP. As shown in Figure 5A, dSHE increased in the VA ECMO group; however, it decreased in the VA ECMO and IABP groups with decreasing VA ECMO flow. The dEEP and dSHE exhibited the same trends. The difference between SHE and EEP indicates the loss of pulsatility from the left ventricle to VA ECMO. IABP changed the trends of dEEP and dSHE, which influenced the loss of pulsatility. dEEP crossed at 2.0 L/min to 1.5 L/min of VA ECMO, which indicated a weak influence of IABP on pulsatility. PPI’ showed an increasing trend with the recovery of cardiac function in both groups (Figure 5B). PPI’ with IABP was twice that without IABP. IABP contributed to an increase in arterial pressure pulsatility.

Figure 6 shows the intersection of the flow in the left ventricle and VA ECMO. The flow of the left ventricle is shown in blue, and the flow of the VA ECMO is shown in red. The intersection of two opposing blood molecules was generated in the aortic arch region during VA ECMO. With the recovery of cardiac function, the intersection moved towards the descending aorta. From 2.5 L/min to 1 L/min for VA ECMO, the left ventricle flow influenced the perfusion of the brain. In other words, the pulmonary function of the oxygen supply needs to be monitored to prevent hypoxia in the brain. In contrast to VA ECMO, blood intersection appeared from the descending aorta to the renal artery with VA ECMO and IABP support, which influenced renal perfusion.

Figure 7 shows the WSS distribution. T1 is the maximum value of blood velocity during the systolic phase, and T2 is the maximum volume of the IABP during the diastolic phase. For VA ECMO, the high region of the WSS was concentrated in the arteria femoralis, especially at the outlet of the catheter. A high region of WSS was seen between the renal artery and arteria femoralis in the VA ECMO and IABP groups. Meanwhile, the WSS increased at the time point of T2, which indicated the effect of the IABP on the vessel. Subsequently, the time-averaged wall shear stress (TAWSS) was calculated. The maximum TAWSS of the renal artery showed significant differences with or without IABP (VA ECMO: 2.02 vs. 1.98 vs. 2.37 vs. 2.61 vs. 2.86 Pa; VA ECMO and IABP: 8.02 vs. 6.99 vs. 6.62 vs. 6.30 vs. 5.83 Pa). An increase of more than twice the maximum TAWSS indicated that the IABP affected the vascular wall of the renal artery.

## 4. Discussion

Despite all the advances in device technology over the past decades, the ECMO system remains non-pulsatile with physiological properties [34]. Although there are some instances of pulsatile ECMO using a specific pulsatile pump, evidence that proves its efficiency is inadequate [35,36]. Therefore, this study demonstrated a finite-element model of the aorta and its main branches under VA ECMO and IABP to explain the relationship between pulsatile blood flow and flow distribution in the circulatory system.

Current VA ECMO systems generally provide continuous flow with a centrifugal pump, which contradicts the weak pulsatile flow of the native heart and weakens systemic pulsatility. Along with cardiac support, VA ECMO increases left ventricular afterload [37,38]. Under these circumstances, the IABP works as a source of pulsatile flow that could reduce LV afterloads and provide other benefits. A series of retrospective studies and meta-analyses have shown a relatively higher survival rate with ECMO plus IABP than that with ECMO alone [29], with a higher weaning rate and other outcomes [39]. IABP is also an independent predictor of in-hospital mortality and complications [40]. Moreover, improved hemodynamics have been reported in other studies when IABP was combined with VA ECMO [41,42]. In our numerical model, ECMO plus IABP provided a higher total blood flow than ECMO alone (Figure 3). In the analysis of arteria femoralis blood flow (Figure 4), IABP increased the systemic flow, including that of the right arteria femoralis (RAF). Meanwhile, the flow of the left arteria femoralis (LAF) was reduced, which could be a response to IABP-induced counter pulsation. These results are in accordance with clinical practices.

The pulsatile property of VA ECMO support has attracted much attention, but the role of IABP has been controversial. Chen et al. reported that concurrent initiation of IABP with ECMO provided better short-term survival and reduced peripheral perfusion complications for post-cardiotomy CS. Hideshi et al. found that pulsatile ECMO increased hemodynamic energy and systemic microcirculation more than non-pulsatile flow in an acute cardiac failure model in piglets [43]. In our study, three parameters, dEEP, dSHE, and PPI, were obtained to estimate pulsatile properties (Figure 5). dEEP and dSHE represent the difference in the values between LV and VA ECMO, which demonstrated the loss of pulsatility. The dSHE of ECMO plus IABP outperformed that of ECMO alone when the blood flow was greater than 2.0 L/min. In other words, IABP induced a loss of pulsatile energy compared to the native heart when the blood flow was greater than 2.0 L/min. The cross of the dEEP curve from 2.0 L/min to 1.5 L/min of VA ECMO demonstrated the weak effect of IABP on pulsatility. ECMO plus IABP and ECMO showed an accordant trend in PPI. PPI is the easiest way to evaluate pulsatile properties, which may be superior to pulse pressure in evaluating cardiovascular outcomes [44]. In our study, the recovery of cardiac function and IABP contributed to an increase in pulsatility of arterial pressure.

Pulsatile flow leads to better tissue perfusion and improves microcirculation. Recently, direct visualization and quantification of microvascular function were used to compare pulsatile and non-pulsatile perfusion, for instance, orthogonal polarization spectral imaging, near-infrared spectroscopy, and vascular occlusion testing. The benefits of pulsatile perfusion can be specifically seen in the maintenance of normal microvascular perfusion and vasoreactivity during and after CPB [45,46]. Further, the improvement of tissue perfusion was specifically seen in the kidney [47,48]. Worsened kidney function has been reported to be a risk factor in ECMO patients [49,50,51,52]. VA ECMO and IABP have been reported to reduce the use of continuous renal replacement therapy. Charlotte et al. found that in cases of isolated kidney perfusion, the pulsatile perfusion mode can achieve better renal perfusion and function preservation [53]. Dhawan et al. reported that abnormal WSS leads to an altered endothelial phenotype and cell signaling [54]. In our analysis, IABP improved cardiac output from 0.78 L/min to 1.43 L/min and facilitated the ejection capacity of the left ventricle (Figure 3 and Figure 6). When the ECMO flow is set between 1.5 L/min and 1.0 L/min, the blood from the left ventricle could reach the renal artery, which promises renal perfusion. In addition, an increase in WSS during the diastole period and twice the maximum TAWSS indicates that IABP affects the vascular wall of the renal artery (Figure 7). Therefore, to preserve the function of the renal artery, a balance between renal perfusion and damage to endothelial cells by WSS is required.

A limitation of this study is that the effect of an IABP inpatient on VA-ECMO includes decreasing afterload, augmenting coronary perfusion, and increasing pulsatility to the end organs. The simulation numerical study cannot distinguish among those effects. Our study calculated the pulsatility parameters and focused on the pulsatile effect produced by IABP. Further animal studies or clinical trials can focus more on other effects of IABP. Besides, we evaluated the hemodynamic effect of pulsatile blood flow in improving the blood flow distribution of VA ECMO with only the mode of arteria femoralis, without grouping it according to pulsatility. The cannulation strategy is limited to peripheral cannulation via the femoral artery, whereas many patients require central cannulation in clinical practice. In central cannulation, venous blood is usually drained from the right atrium, and an arterial cannula is inserted into the ascending aorta. Therefore, in the future, we will consider building a model with central cannulation.

## 5. Conclusions

Finite-element models consisting of the aorta, VA ECMO, and IABP are proposed for fluid-structure interaction calculation of the mechanical response. With the recovery of the left ventricle and the flow decrease of VA ECMO, the effective blood of IABP increases. The difference between SHE and EEP indicates the loss of pulsatility from the left ventricle to VA ECMO. dEEP crosses at 2.0 L/min to 1.5 L/min of VA ECMO, showing the similarity of hemodynamic energy loss with the weak influence of IABP.

## Figures and Tables

**Figure 1 bioengineering-09-00487-f001:**
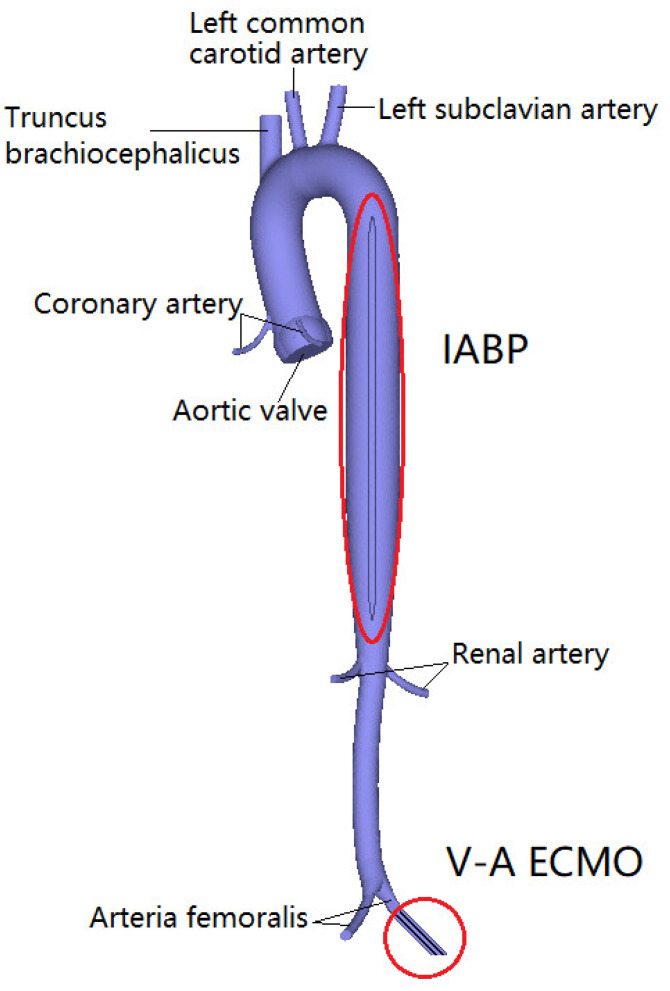
The geometric model of aortic, V-A ECMO, and IABP.

**Figure 2 bioengineering-09-00487-f002:**
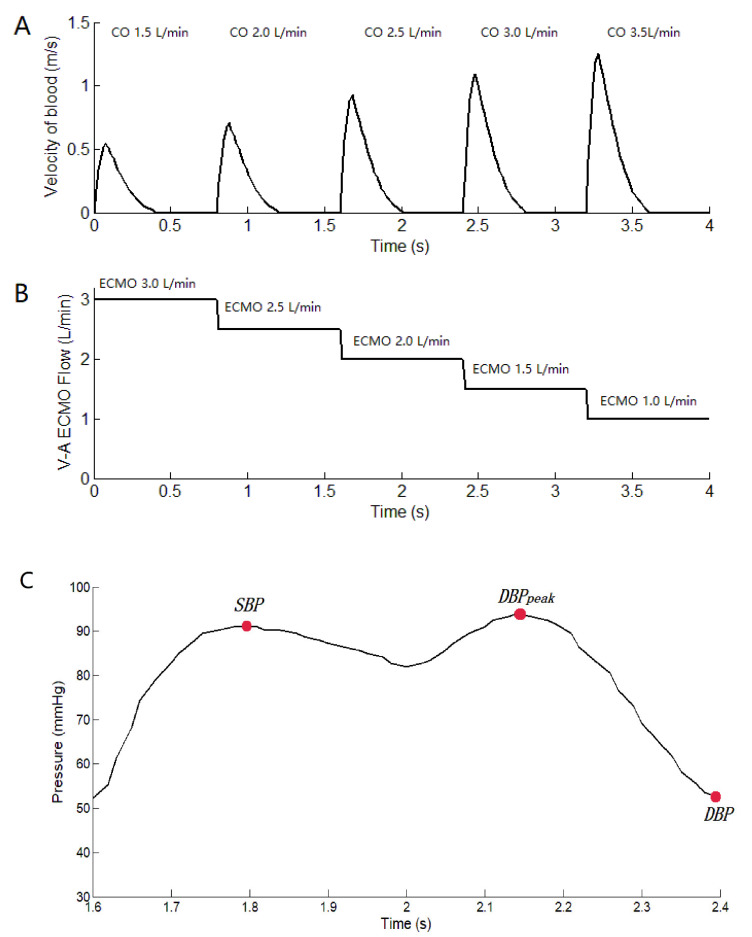
The flow velocity of the aortic valve and the flow of V-A ECMO. (**A**): the flow velocity of the aortic valve from the echocardiographs. CO is calculated by the formula $\mathrm{CO}=\int Vdt*A_{av}*HR$ where *V* is the flow velocity of the aortic valve. *A*_av_ is the area of the aortic valve (715.44 mm^2^), and HR is the heart rate (75 bpm). (**B**): the flow of V-A ECMO, which is setting value. (**C**) The curve of arterial pressure with IABP support, which is marked with SBP, DBPpeak, DBP.

**Figure 3 bioengineering-09-00487-f003:**
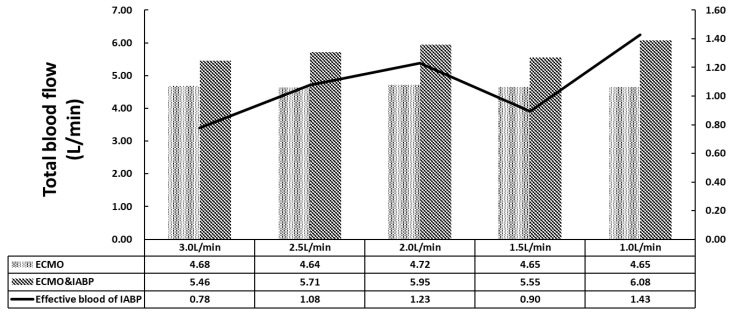
The total blood flow of coronary artery, brain, renal and lower limbs.

**Figure 4 bioengineering-09-00487-f004:**
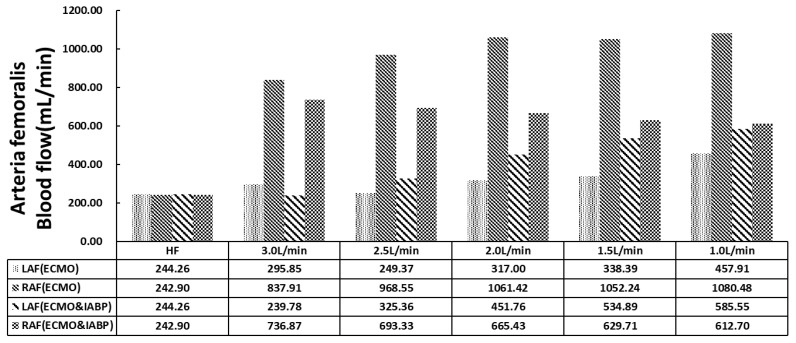
The histogram of arteria femoralis flow.

**Figure 5 bioengineering-09-00487-f005:**
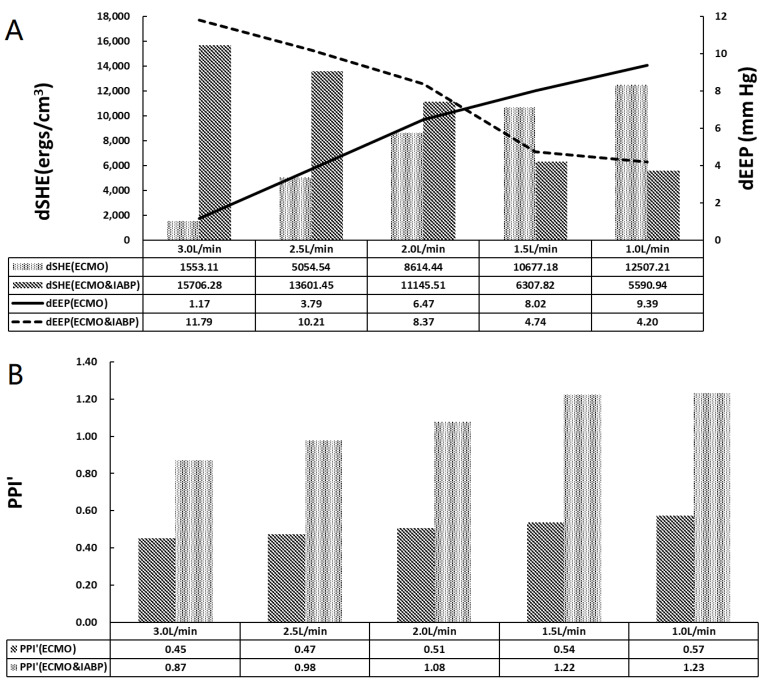
The pulsatile properties of arterial. (**A**) The histogram of dSHE and the curve of dEEP, dEEP is the difference value between LV EEP and V-A ECMO EEP; (**B**) The histogram of PPI’.

**Figure 6 bioengineering-09-00487-f006:**
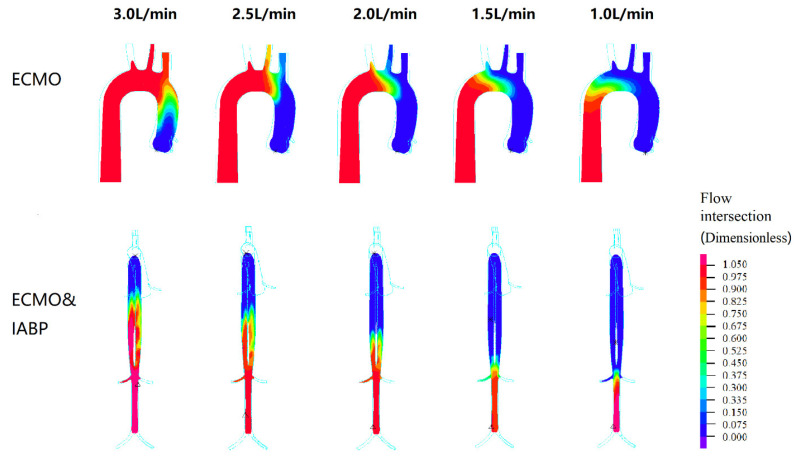
The flow intersection as a dimensionless parameter during left ventricular and V-A ECMO. The flow of the left ventricle is shown in blue, and the flow of the VA ECMO is shown in red.

**Figure 7 bioengineering-09-00487-f007:**
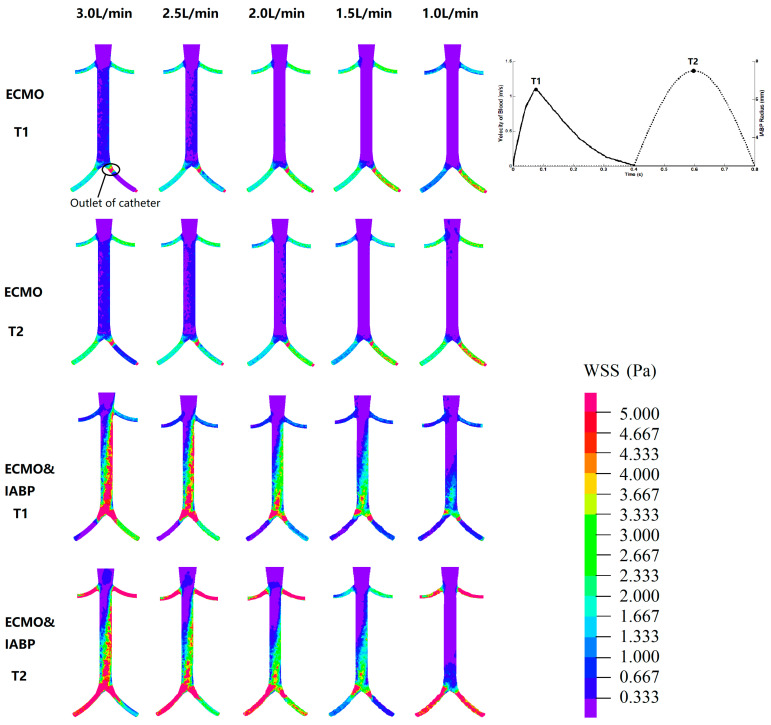
The WSS distribution. T1 is the maximum value of blood velocity at the systole period; T2 is the maximum volume of IABP at the diastole period.

**Table 1 bioengineering-09-00487-t001:** The group of numerical analyses.

Group A	Group B		Group C		
CO (L/min)	CO (L/min)	ECMO (L/min)	CO (L/min)	ECMO (L/min)	IABP (mL)
1.5	1.5	3.0	1.5	3.0	30
	2.0	2.5	2.0	2.5	
	2.5	2.0	2.5	2.0	
	3.0	1.5	3.0	1.5	
	3.5	1.0	3.5	1.0	

## Data Availability

The study does not report any data.

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
