# Peer review of "Hemodynamic Effect of Pulsatile on Blood Flow Distribution with VA ECMO: A Numerical Study"

_bioengineering, 2022, doi:10.3390/bioengineering9100487_

Round 1
Reviewer 1 Report
I appreciate the opportunity to review this interesting manuscript designed to address a topic that is clinically relevant to use of mechanical circulatory support devices in patients with cardiogenic shock. If i understand them correctly, the authors aim to evaluate the impact of the addition of an IABP to a finite-element model of VA-ECMO.
One technical point before my thoughts below:
- in the abstract, line 24. An abbreviation PPI' presents itself without an explanation. The author's intention here is unclear.
The clinical relevance of the topic should not be understated; however I have a few questions. First, in Group A (designated as cardiogenic shock with 1.5 lpm cardiac output); how is that cardiac output generated? Is it just through the ECMO circuit or is it ejected into the aorta by a separate pump? Is it simulated? If so, what is the heart rate or stroke volume? What is the blood pressure from this "native" flow? Perhaps just a few sentences to explain this would be helpful. For example, table 1 uses varying amounts of "CO" and varying amounts of "ECMO", so it might help to understand the direction of this output. Furthermore, i wound expect the addition of an IABP to increase the "native" cardiac output by a certain amount (most suggest at least 0.5 lpm). Is it possible that the differences reported later are in fact more related to an increase in the total cardiac output rather than simply the pressure changes produced by the pulse?
From my perspective, one of the limitations with this analysis is that the effect of an IABP in patient on VA-ECMO is actually much more complex that simply the generation of "a pulse". Is its value in decrease afterload sufficiently to allow the aortic valve to open in a situation where it otherwise would not? Is its value in augmenting coronary perfusion to increase contractility in the native heart. Or is its value in the increased pulsatility to the end organs?
Now, it is possible that the generation of that pulse is clinically relevant, especially at the organ level; however, it has also been argued that because most of the blood flow at the capillary level is in fact continuous flow, the "pulse" is probably unnecessary.
Personally, I would recommend spending some additional time on these topics in the discussion.
Likewise, I would recommend making the conclusions more concise. What is the primary take-home point? If it truly is that "with the recovery of the left ventricle and flow decrease of VA-ECMO, the effective blood of IABP increases."; then I am not sure that is meaningful. The three or four sentences that follow, in my opinion, only serve to confuse the reader more. I would recommend revising the conclusion completely.
Reviewer 2 Report
This study analyzes the hemodynamic effects of pulsatile flow on V-A ECMO supported patient, including blood flow distribution, arterial pulsatile properties and WSS distribution. The results show that the implantation of IABP is beneficial to increase systemic blood flow, balance the blood distribution between right and left femoral arteries, and increase the pulsatility of arterial pressure, but it also causes greater damage to renal arteries. This article is interesting but following problems need to be addressed:
1. Please hide the coordinate system in Figure 1 to avoid misunderstanding and enlarge the left femoral artery section to show the ECMO cannula clearly.
2. L164 “The Energy Equivalent Pressure” should be “The Equivalent Energy Pressure”
3. Please explain Equation 6 and Equation 7 further and draw the arterial pressure waveform with IABP support to distinguish DBPpeak from DBP.
4. L228 “A: The histogram of SHE and the curve of dEEP,” should be “A: The histogram of dSHE and the curve of dEEP,”
5. The time point, physical quantity and unit corresponding to Fig. 6 are not indicated.
6. The unit in figure 7 is not indicated.
7. In this manuscript, only WSS is used to analyze vascular wall damage, and it is recommended to combine WSS with OSI for a more comprehensive analysis.
8. It can be seen from Fig. 4 that the blood flow of LAF increased after IABP implantation while that of RAF decreased, which is contrary to the description of L273 to L274.
Round 2
Reviewer 2 Report
The authors revised the manuscirpt very carefully and I am satified to the responses.